# Protocol for surveillance of antimicrobial-resistant bacteria causing community-acquired urinary tract infections in low-income countries

Mtebe Venance Majigo[1,2]*, Stephen Mshana[1,3], Erick Komba[1,4], Nyambura Moremi[1,5], Mecky Matee[1,2]

1 SACIDS Foundation for One Health, Dar es Salaam, Tanzania, 2 Muhimbili University of Health and Allied Sciences, Dar es Salaam, Tanzania, 3 Catholic University of Health and Allied Sciences, Dar es Salaam, Tanzania, 4 Sokoine University of Agriculture, Morogoro, Tanzania, 5 National Health Laboratory Quality Assurance and Training Centre, Dar es Salaam, Tanzania

* mmajigo@gmail.com

**Data Availability Statement:** No datasets were generated or analysed during the current study.

## Abstract

The spread of drug-resistant bacteria into the community is an urgent threat. In most low-middle-income countries (LMICs) settings, community-acquired infection (CAI) is empirically treated with no data to support the choice of antibiotics, hence contributing to resistance development. Continuous antimicrobial resistance (AMR) data on community-acquired pathogens are needed to draft empirical treatment guidelines, especially for areas with limited culture and susceptibility testing. Despite the importance of addressing antibiotic-resistant pathogens in the community setting, protocols for the surveillance of AMR bacterial infections are lacking in most (LMICs). We present a protocol for surveillance of AMR in LMICs using urinary tract infection (UTI) as a proxy for CAI to enable users to quantify and establish the drivers of AMR bacteria causing UTI. The protocol intends to assist users in designing a sustainable surveillance program for AMR in the community involving children above two years of age and adults presenting to a primary health facility for healthcare. Implementation of the protocol requires initial preparation of the laboratories to be involved, surveillance areas, selection of priority bacteria and antimicrobials to be used, and the design of a coordinated sampling plan. Recruitment should occur continuously in selected health facilities for at least 12 months to observe seasonal trends of AMR. At least 10 mL of clean-catch mid-stream urine must be collected into 20 mL calibrated sterile screw-capped universal bottles lined with 0.2 mg boric acid and transported to the testing laboratory. Utilise the data system that generates standard reports for patient care to be shared internally and externally in the regions and the world through global platforms such as the Global Antimicrobial Resistance Surveillance System.

**Funding:** The Fleming Fund Round 2 Regional Grant number FF49/440 funded the development of this protocol. The funder had no involvement in the design and development of the protocol.

**Competing interests:** The authors have declared that no competing interests exist.

## Introduction

Antimicrobial resistance (AMR) is a global public health crisis that impacts livelihoods, food security, and development, particularly in low-middle-income countries (LMIC). The spread of drug-resistant bacteria into the community is an urgent threat associated with increased morbidity, mortality, healthcare costs, and antibiotic use [1–3]. Among drug-resistant pathogens, extended-spectrum β-lactamase producing Enterobacterales (ESBL-PE) and multidrug-resistant to non-β-lactam antibiotics are currently the most critical threats to public health [4–6]. ESBL-PE strains are clinically significant because they limit the effectiveness of cephalosporins, one of the most commonly used antibiotics [7]. Delay recognition and inappropriate treatment of infections caused by multidrug-resistant (MDR) pathogens, such as ESBL-PE, have increased mortality [2, 8].

Urinary tract infection (UTI) leads to significant morbidity and health care expenditures, particularly in LMIC [9]. The predominant pathogens causing UTI are gram-negative bacteria from the Enterobacterales group. These pathogens include Uropathogenic *E. coli*, *Klebsiella pneumoniae* complex, *Proteus* spp., and *Enterobacter* spp [10]. The World Health Organization (WHO) has classified Enterobacterales that are resistant to carbapenems and third-generation cephalosporins as critical priority pathogens. This classification necessitates the development of research and development strategies for new antibiotics [10].

In most LMIC settings, UTI is empirically treated with no data to support the choice of antibiotics, hence contributing to resistance development. Continuous AMR data on community-acquired pathogens are needed to draft empirical treatment guidelines, especially for areas with limited culture and susceptibility testing [11]. With such data, policymakers can design appropriate AMR control strategies that help guide community practitioners. [12] Surveillance data on UTI can assist in understanding community-level magnitude, drivers, and transmission dynamics.

Despite the importance of addressing antibiotic-resistant pathogens in the community setting, protocols for the surveillance of AMR bacterial infections are lacking in most LMICs [13, 14]. Access to data about AMR drivers in community settings is necessary for governments and scientists to make evidence-based policy and intervention decisions. The protocol is required to assist users in designing a sustainable surveillance program for AMR and Community-Acquired Infections (CAI) [12].

We present a protocol for surveillance of AMR in low-income countries using UTI as a proxy for CAI to enable users to quantify and establish the drivers of community-acquired AMR bacteria causing UTI. We conducted a cross-sectional health center-based survey as a pilot for the protocol. The survey was conducted in Tanzania, Mwanza, and Dar es Salaam regions for five months from July to November 2021. The findings from the survey provide evidence that the protocol works [15, 16]. In addition, the protocol aligns with WHO AMR surveillance guidelines [17].

## Materials and methods

The material and methods for implementing the protocol for surveillance of antimicrobial-resistant bacteria causing community-acquired urinary tract infections in low-middle-income countries are published on protocols.io. https://protocols.io/view/survsilence-of-antimicrobial-resistant-bacteria-c-c7ryzm7w. The protocol is included for printing as a supporting information file (S1 File) with this article.

## Ethics declarations

The National Institute for Medical Research (NIMR) cleared the study with a certificate numbered NIMR/HQ/R.8a/ Vol.IX/3580 of 16 December 2020, which resulted in the development

of this protocol. In implementing this protocol, ethics approval must be sought according to national guidelines; confidentiality of patients' data should be ensured.

## Expected results

Implementation of this protocol is expected to give the following results: 1) Strengthen the existing AMR surveillance system, including epidemiological skills (sample design, data analysis, and reporting), sample collection and processing, laboratory diagnostic capacity, and data management and analysis. 2) Generate baseline estimates of the prevalence of AMR bacteria causing community-acquired UTI, of which future surveillance rounds can be compared to identify new strains, patterns, and trends in prevalence and dispersion. 3) Document the diversity of phenotypes of uropathogens based on the susceptibility patterns in the community. 4) Identify the primary drivers of community-acquired AMR pathogens causing UTI. 5) Inform priorities and evolution of the AMR surveillance system to improve understanding of the prevalence and consequences of AMR in the community.

The protocol is designed to be adopted by local governments and integrated into existing national AMR surveillance systems. Ongoing AMR surveillance is crucial for tracking resistance patterns. This surveillance provides essential data to inform policy and decision-making, helping to mitigate the consequences of identified AMR.

This protocol outlines how to conduct alert organism tracking for MDR bacteria causing community-acquired UTIs and is complementary to routine surveillance. Alert organism tracking is generally not the first step of a surveillance system but entails the addition and prioritization of particular pathogens of public health significance. Alerts will automatically be generated by the WHONET software following data entry due to pre-defined algorithms fed into the software.

## Supporting information

**S1 File. Step-by-step protocol for conducting surveillance of antimicrobial-resistant bacteria causing community-acquired urinary tract infections.** The protocol is also available on protocols.io: dx.doi.org/10.17504/protocols.io.kqdg3xdneg25/.
(PDF)

## Acknowledgments

We thank the Ending Pandemics program, Connecting Organizations for Regional Disease Surveillance (CORDS), and East and Southern Africa regional Technical Working Group members for providing valuable insight. We are grateful for the guidance provided by the Ending Pandemics Team. We gratefully acknowledge the technical support from the Mott MacDonald team for developing this protocol.

## Author Contributions

**Conceptualization:** Mtebe Venance Majigo, Stephen Mshana, Erick Komba, Nyambura Moremi, Mecky Matee.

**Methodology:** Mtebe Venance Majigo, Stephen Mshana, Erick Komba, Nyambura Moremi, Mecky Matee.

**Supervision:** Stephen Mshana, Mecky Matee.

**Writing – original draft:** Mtebe Venance Majigo.

**Writing – review & editing:** Mtebe Venance Majigo, Stephen Mshana, Erick Komba, Nyambura Moremi, Mecky Matee.

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
