## [Decision Letter · Decision Letter 0]

7 Mar 2024

PONE-D-23-38872Protocol for surveillance of antimicrobial-resistant bacteria causing community-acquired urinary tract infections in low-income countriesPLOS ONE

Dear Dr. Majigo,

Thank you for submitting your manuscript to PLOS ONE. After careful consideration, we feel that it has merit but does not fully meet PLOS ONE’s publication criteria as it currently stands. Therefore, we invite you to submit a revised version of the manuscript that addresses the points raised during the review process.

We look forward to receiving your revised manuscript.

Kind regards,

Mabel Kamweli Aworh, DVM, MPH, PhD. FCVSN

Academic Editor

PLOS ONE

Journal Requirements:

"We thank the Ending Pandemics program, Connecting Organizations for Regional Disease Surveillance (CORDS), and East and Southern Africa regional Technical Working Group members for providing valuable insight. We are grateful for the guidance provided by the Ending Pandemics Team. We gratefully acknowledge the technical and funding support received for developing this protocol from the Fleming Fund Grant provided by the UK Department of Health and Social Care (DHSC) and the Mott MacDonald agent for administering this grant." 

"The Fleming Fund Regional grant number FF49/440, Regional Grants Round 2, funded the development of this protocol. The funder had no involvement in the design and development of the protocol."

4. When you submit your revision, please provide a PDF version of your protocol as generated by protocols.io (the file will have the protocols.io logo in the upper right corner of the first page) as a Supporting Information file. The filename should be S1_file.pdf, and you should enter “S1 File” into the Description field. Any additional protocols should be numbered S2, S3, and so on. 

If you have not yet uploaded your protocol to protocols.io, you are invited to use the platform’s protocol entry service [https://www.protocols.io/we-enter-protocols] for doing so, at no charge. Through this service, the team at protocols.io will enter your protocol for you and format it in a way that takes advantage of the platform’s features. When submitting your protocol to the protocol entry service please include the customer code PLOS2022 in the Note field and indicate that your protocol is associated with a PLOS ONE Lab Protocol Submission. You should also include the title and manuscript number of your PLOS ONE submission.

Reviewers' comments:

Reviewer's Responses to Questions

**Comments to the Author**

1. Does the manuscript report a protocol which is of utility to the research community and adds value to the published literature?

Reviewer #1: Yes

Reviewer #2: Yes

2. Has the protocol been described in sufficient detail?

To answer this question, please click the link to protocols.io in the Materials and Methods section of the manuscript (if a link has been provided) or consult the step-by-step protocol in the Supporting Information files.

The step-by-step protocol should contain sufficient detail for another researcher to be able to reproduce all experiments and analyses.

Reviewer #1: No

Reviewer #2: No

3. Does the protocol describe a validated method?

Reviewer #1: Yes

Reviewer #2: Yes

4. If the manuscript contains new data, have the authors made this data fully available?

Reviewer #1: N/A

Reviewer #2: N/A

**5. Is the article presented in an intelligible fashion and written in standard English?**

Reviewer #1: Yes

Reviewer #2: **No: **The manuscript will benefit from garamma check

6. Review Comments to the Author

Reviewer #1: Good to see such critical thinking on ways to improve AMR surveillance, and to reduce its burden on the health care system.

I have comments attched.

Reviewer #2: This document outlines a well-structured protocol for antimicrobial resistance (AMR) surveillance of community-acquired urinary tract infections (UTI). The document clearly defines target populations, enrollment criteria, and surveillance areas. It emphasizes the importance of standardized operating procedures (SOPs) for sample collection, handling, transportation, and laboratory testing. The protocol mentions proficiency testing and internal quality assurance for maintaining data integrity. It highlights the importance of data management, analysis, reporting, and sharing.

There are a few areas needing clarification as follows;

1. WHO has classified Enterobacterales carbapenem-resistant and 3rd generation cephalosporin-resistant as critical priority pathogens – This statement is unclear

2. who are residents of a given surveillance area and passively presenting to a health facility for health care – please clarify what ‘passively’ means in this case

3. In addition, the protocol aligns with WHO AMR surveillance guidelines – Reference required

4. community-acquired urinary tract infections in low-income countries are published on protocols.io. – Do the writers mean low income or low-middle income?

5. Table 1 – Lab tests such ass urinalysis and urine microscopy are not usually regarded as ‘signs’

6. Table 3 – May be rather simplistic as it does not into account intrinsic resistance

7. For urinalysis- Include any caution in specimen preserved with boric acid

8. Isolates should be safely transported monthly to the AMR Reference laboratory for additional tests and storage at -80 OC per country SOP for isolate management – is storage at this temperature possible at district level?

9. The QA for AST does not comply with CLSI as stated in the protocol

Overall, a more comprehensive protocol will allow for seamless implementation.

7. PLOS authors have the option to publish the peer review history of their article (what does this mean?). If published, this will include your full peer review and any attached files.

Reviewer #1: No

Reviewer #2: **Yes: **Nubwa Medugu

---

## [Author Response · Author response to Decision Letter 0]

28 Apr 2024

I have attached the response to reviewers

---

## [Decision Letter · Decision Letter 1]

13 May 2024

Protocol for surveillance of antimicrobial-resistant bacteria causing community-acquired urinary tract infections in low-income countries

PONE-D-23-38872R1

Dear Dr. Majigo

We’re pleased to inform you that your manuscript has been judged scientifically suitable for publication and will be formally accepted for publication once it meets all outstanding technical requirements.

Kind regards,

Mabel Kamweli Aworh, DVM, MPH, PhD. FCVSN

Academic Editor

PLOS ONE

Additional Editor Comments (optional):

Reviewers' comments:

Reviewer's Responses to Questions

**Comments to the Author**

1. Does the manuscript report a protocol which is of utility to the research community and adds value to the published literature?

Reviewer #1: Yes

Reviewer #2: Yes

2. Has the protocol been described in sufficient detail?

To answer this question, please click the link to protocols.io in the Materials and Methods section of the manuscript (if a link has been provided) or consult the step-by-step protocol in the Supporting Information files.

The step-by-step protocol should contain sufficient detail for another researcher to be able to reproduce all experiments and analyses.

Reviewer #1: Yes

Reviewer #2: Yes

3. Does the protocol describe a validated method?

Reviewer #1: Yes

Reviewer #2: Yes

4. If the manuscript contains new data, have the authors made this data fully available?

Reviewer #1: N/A

Reviewer #2: N/A

**5. Is the article presented in an intelligible fashion and written in standard English?**

Reviewer #1: Yes

Reviewer #2: Yes

6. Review Comments to the Author

Reviewer #1: The author made all the corrections that were identified in the last version of the manuscript.

The author should align the references according to the journal standards.

Reviewer #2: I am pleased to recommend acceptance of the revised manuscript titled "Protocol for Surveillance of Antimicrobial-Resistant Bacteria Causing Community-Acquired Urinary Tract Infections in Low-Income Countries." The authors have comprehensively addressed all feedback and suggestions from the reviewers. The protocol offers a clear and well-defined approach for monitoring antimicrobial resistance (AMR) in this critical area for low-income countries.

This study holds significant value as it directly addresses the growing threat of AMR in community-acquired UTIs, a prevalent health concern in these settings. The outlined protocol provides a practical and sustainable approach for resource-limited settings, making it widely applicable.

Given the thorough revisions, the absence of ethical or technical concerns, and the manuscript's potential impact, I strongly recommend acceptance for publication.

7. PLOS authors have the option to publish the peer review history of their article (what does this mean?). If published, this will include your full peer review and any attached files.

Reviewer #1: No

Reviewer #2: **Yes: **Dr Nubwa Medugu

---

## [Editor Report · Acceptance letter]

20 May 2024

PONE-D-23-38872R1 

PLOS ONE

Dear Dr. Majigo, 

I'm pleased to inform you that your manuscript has been deemed suitable for publication in PLOS ONE. Congratulations! Your manuscript is now being handed over to our production team.

Kind regards, 

on behalf of

Dr. Mabel Kamweli Aworh 

Academic Editor

PLOS ONE